# Thyroid Cancer Detection in a Routine Clinical Setting: Performance of ACR TI-RADS, FNAC, and Molecular Testing in Prospective Cohort Study

**DOI:** 10.3390/biomedicines10050954

**Published:** 2022-04-20

**Authors:** Tereza Grimmichova, Petra Pacesova, Martin Hill, Barbora Pekova, Marketa Vankova, Jitka Moravcova, Jana Vrbikova, Zdenek Novak, Karolina Mastnikova, Eliska Vaclavikova, Josef Vcelak, Bela Bendlova, Jana Drozenova, Vlasta Sykorova

**Affiliations:** 1Institute of Endocrinology, Narodni 8, 11694 Prague, Czech Republic; ppacesova@endo.cz (P.P.); mhill@endo.cz (M.H.); bpekova@endo.cz (B.P.); mvankova@endo.cz (M.V.); jmoravcova@endo.cz (J.M.); jvrbikova@endo.cz (J.V.); znovak@endo.cz (Z.N.); kmastnikova@endo.cz (K.M.); evaclavikova@endo.cz (E.V.); jvcelak@endo.cz (J.V.); bbendlova@endo.cz (B.B.); vsykorova@endo.cz (V.S.); 2Internal Clinic, University Hospital Kralovske Vinohrady, Third Faculty of Medicine, Charles University, Srobarova 1150/50, 10034 Prague, Czech Republic; 3Department of Pathology, University Hospital Kralovske Vinohrady, Third Faculty of Medicine, Charles University, Srobarova 1150/50, 10034 Prague, Czech Republic; drozenovaj@fnkv.cz

**Keywords:** thyroid nodule, thyroid cancer, ACR-TIRADS, FNAC, molecular testing, fusions, *BRAF*, *TERT*, *RAS*

## Abstract

The aim of our study was to address the potential for improvements in thyroid cancer detection in routine clinical settings using a clinical examination, the American College of Radiology Thyroid Imaging Reporting and Database System (ACR TI-RADS), and fine-needle aspiration cytology (FNAC) concurrently with molecular diagnostics. A prospective cohort study was performed on 178 patients. DNA from FNA samples was used for next-generation sequencing to identify mutations in the genes *BRAF*, *HRAS*, *KRAS*, *NRAS*, and *TERT*. RNA was used for real-time PCR to detect fusion genes. The strongest relevant positive predictors for malignancy were the presence of genetic mutations (*p* < 0.01), followed by FNAC (*p* < 0.01) and ACR TI-RADS (*p* < 0.01). Overall, FNAC, ACR TI-RADS, and genetic testing reached a sensitivity of up to 96.1% and a specificity of 88.3%, with a diagnostic odds ratio (DOR) of 183.6. Sensitivity, specificity, and DOR decreased to 75.0%, 88.9%, and 24.0, respectively, for indeterminate (Bethesda III, IV) FNAC results. FNA molecular testing has substantial potential for thyroid malignancy detection and could lead to improvements in our approaches to patients. However, clinical examination, ACR TI-RADS, and FNAC remained relevant factors.

## 1. Introduction

Thyroid nodules are prevalent in the general population with a malignant rate of 5–15% [1]. It is essential to distinguish between benign and malignant thyroid nodules. Ultrasonography (US) is the initial modality for the evaluation and workup of thyroid nodules. A thyroid imaging report and data system (TIRADS) has been proposed for the classification and malignant risk-stratification of thyroid nodules, including minimizing the interobserver variability [2,3,4]. Fine-needle aspiration cytology (FNAC) is considered the most efficient method for evaluating thyroid lesions. However, the cytology results may be indeterminate (Bethesda III, IV) in up to 25% of thyroid nodules, and the sensitivity of FNAC decreases with nodule size [5,6]. The use of diagnostic molecular markers (to “rule out” or “rule in” the presence of thyroid malignancy) has been proposed, especially in indeterminate thyroid FNAC specimens, with the aim to inform decision making on primary surgical treatments (i.e., the decision to perform surgery and if so, the extent of surgery). An ideal “rule-in” test would have a positive predictive value (PPV) for a proven malignancy determined by histology resembling a malignant cytologic diagnosis (98.6%), and an ideal “rule-out” test would have a negative predictive value (NPV) like a benign cytologic diagnosis (96.3%). These estimates have been based on a meta-analysis of the performance of the Bethesda system [7], and these would hold true with a reasonable degree of precision and reproducibility [1].Mutational testing has been intended to be used as a rule-in test because of the relatively high reported specificity (86–100%) and PPV (84–100%). *BRAF V600E* single mutation testing has not been efficient in reliably ruling out the presence of malignancy because sensitivity has been too low despite high specificity of approximately 99%. Therefore, multiple genes must be analyzed in mutational panels [8,9,10,11].

It is also necessary to differentiate between thyroid cancer (TC) with progressive features and tumors with an indolent course [12]. Finally, recommendations of treatment strategies should ideally be tailored to individual patients. In our prospective cohort study, we assessed the performance of TC detection according to (1) basic clinical examination, (2) thyroid nodule stratification by ultrasound using the American College of Radiology Thyroid Imaging Reporting and Database System (ACR TI-RADS), (3) FNAC, and (4) molecular testing. In addition, we evaluated possible changes in our approaches to treatment in routine clinical settings.

## 2. Materials and Methods

### 2.1. Patients and Study Design

In our prospective cohort study, we included 178 patients (143 women and 35 men) in a country with iodine sufficiency [13]. The patients were consecutively recruited from August 2017 to April 2021. Patient history, biochemical testing, ultrasound of the neck (US), fine-needle aspiration biopsy (FNA), and FNA molecular testing were done at the Institute of Endocrinology and Internal clinic, University Hospital Kralovske Vinohrady. The flowchart of the study can be seen in Figure 1. The protocol of this study complied with the Declaration of Helsinki, and before entering the study, written informed consent was obtained from all patients after they received both written and oral information. The study was approved by the ethical committee of the Institute of Endocrinology.

### 2.2. Clinical and Biochemical Characteristics

An “incidental” cancer was defined as an asymptomatic thyroid cancer detected incidentally during a medical imaging test, physical examination, or surgery studies performed for other reasons [14]. The diagnosis of autoimmune thyroid disease (AITD) was given for cases with positive thyroid autoantibodies and/or the hypoechogenic pattern typical for AITD during the US examination. Basal blood samples for the determination of TSH, fT4, fT3, anti-thyroid peroxidase antibodies (anti-TPO), anti-thyroglobulin (anti-Tg), thyrotropin receptor antibodies (TRAbs), and glucose were taken at the same time as FNA was performed. Serum TSH (normal range: 0.27–4.20 mUI/L), fT4 (12.00–22.00 pmol/L), fT3 (3.10–6.80 pmol/L), and TRAbs (0.30–1.75 IU/L) concentrations were measured using the ECLIA method (Cobas 6000, Roche, Manheim, Germany). Serum anti-Tg (0.01–120 IU/mL) and anti-TPO (0.01–40 IU/mL) were analyzed by ELISA (Aeskulisa, Aesku Diagnostics, Wendelsheim, Germany). Spectrophotometrical (UV)-hexokinase method was used to measure glucose (3.9–5.6 mmol/L).

### 2.3. Ultrasound Examination

The ACR TI-RADS ultrasound thyroid nodule guidance has been previously published. Briefly, ACR TI-RADS is a scoring system, with points given for the description of composition, echogenicity, shape, margin, and echogenic foci of thyroid nodules. It consists of five groups: TR1—benign, TR2—not suspicious, TR3—mildly suspicious, TR4—moderately suspicious, and TR5—highly suspicious [15,16]. US was performed at a frequency of 12.5 MHz on a Philips Epiq5.

### 2.4. Fine Needle Aspiration Biopsy and Cytology

FNA was performed in 178 thyroid nodules using the pistol technique under US guidance. In 15 patients, two thyroid nodules were biopsied and/or FNA repeated due to previous Bethesda III results. FNA was generally performed on thyroid nodules sized >1 cm; only three patients had smaller thyroid nodules with highly US suspicious features. FNA was preferentially performed on thyroid nodules with the suspicious US features such as hypoechogenicity, irregular or microlobulated margins, taller-than-wide, punctate echogenic foci, and solid components. FNA was done once for each thyroid nodule using a 20-gauge needle attached to a 20 mL syringe without using local anesthesia. A picture of the thyroid gland pointing out the thyroid nodule undergoing FNA is a part of the uniform reports. Aspirated material was partly expelled onto glass slides and sent for cytopathology examinations. The residual material in the needle and the needle wash were placed into a tube containing 500 μL of RNA later (Sigma-Aldrich, St. Louis, MO, USA) or 600 μL RNA/DNA Shield (Zymo Research, Irvine, CA, USA). The expert cytopathologists evaluated the May-Grünwald/Giemsa, hematoxylin, and eosin-stained specimens. The Bethesda System for Reporting Thyroid Cytopathology (TBSRTC) 2017 was followed with groups of (I) nondiagnostic or unsatisfactory; (II) benign; (III) atypia of undetermined significance (AUS) or follicular lesion of undetermined significance (FLUS); (IV) follicular neoplasm or suspicious for a follicular neoplasm; (V) suspicious for malignancy; and (VI) malignant [17]. All Bethesda I results were excluded from this study. In the group of indeterminate FNAC, results of Bethesda III and IV categories were included.

### 2.5. Genetical Analysis

DNA and RNA from FNA samples stored in RNA later Stabilization Solution (Sigma, St. Louis, MI, USA) were extracted using an AllPrep DNA/RNA/miRNA Universal Kit (Qiagen, Venlo, The Netherlands). The concentrations of samples were measured using a fluorometer (Quantus, Promega, WI, USA). We gradually established testing procedures mainly in samples evaluated as Bethesda categories III and above. First, we analyze DNA for the most common mutation *V600E* in the *BRAF* gene using allele-specific real-time PCR (LC480, Roche, Penzberg, Germany). *BRAF*-positive samples are screened for *TERT* mutations using direct sequencing (CEQ 8000, Beckman Coulter, CA, USA). *BRAF*-negative samples are analyzed by next-generation sequencing using the Thyro-ID panel (MiSeq, Illumina, San Diego, CA, USA) examining other 12 genes (*BRAF*, *HRAS*, *KRAS*, *NRAS*, *TERT*, *AKT1*, *EGFR*, *TP53*, *PTEN*, *PIK3CA*, *CDKN2A*, *NOTCH*, and *CTNNB1)*. The samples negative in the NGS panel are subjected to detection of 23 fusion genes, including ALK, BRAF, GLIS, NTRK1, NTRK3, PPARG, and RET genes. The range of molecular genetic testing was expanded throughout the study. First, only a *BRAF V600E* mutation analysis in 60 samples was performed. Furthermore, 101 samples were also prepared using the Thyro-ID panel (4bases, Manno, Switzerland). Six samples were prepared for sequencing using the VariantPlex Comprehensive Thyroid and Lung panel (ArcherDx, Boulder, CO, USA). The VariantPlex panel includes the following genes: *BRAF*, *HRAS*, *KRAS*, *NRAS*, *TERT*, *AKT1*, *EGFR*, *TP53*, *PTEN*, *PIK3CA*, *CTNNB1*, *ALK*, *CCND1*, *DDR2*, *EIF1AX*, *ERBB2*, *FGFR1–3*, *GNAS*, *IDH1–2*, *KIT*, *MAP2K1*, *MDM2*, *MET*, *PDGFRA*, *RET*, *ROS1*, *STK11*, and *TSHR*. The detection of fusion genes was performed by real-time PCR as described previously [18]. First, only the most common *RET/PTC1*, *ETV6/NTRK3,* and *RET/PTC3* in 50 samples were analyzed. Later, the analyses of fusion genes expanded to include another four fusion genes (*STRN/ALK*, *TPM3/NTRK1*, *PAX8/PPARY*, *SQSTM1/NTRK3*), and 24 samples were tested for them. Due to the gradual expansion of tested genes, in four samples, the fusion gene was later identified in postoperative specimens (2× *ETV6/NTRK3*, 1× *SQSTM1/NTRK1*, 1× *EML4/ALK*) using the FusionPlexComprehensive Thyroid and Lung panel (ArcherDx, Boulder, CO, USA).

Histopathology of surgical specimens was performed in 178 patients and was reviewed by expert thyroid pathologists who were mostly blinded to molecular test results.

### 2.6. Statistical Analysis

Statistical significance was set for *p*-values < 0.05. Sensitivity, specificity, the positive likelihood ratio (LR+), and the negative likelihood ratio (LR-) of each test were calculated with 95% confidence intervals. ROC analysis was used to determine cut-off values for particular parameters. Before the application of parametric methods, the original metric data were transformed by power transformations to approximate the Gaussian distribution and constant variance. The homogeneity and symmetry of transformed data were checked, as shown in our previous papers [19]. Statistical software Statgraphics Centurion 18 from Statpoint (The Plains, VA, USA) was used for power transformations and parametric ANOVA. Kruskal–Wallis nonparametric one-way ANOVA was used for comparison of clinical, biochemical, and imaging characteristics between B (benign), M (malignant), and MB (borderline tumor) cohorts of patients. Multivariate regression with reduction of dimensionality (orthogonal projection into a latent structure, OPLS) was used to discriminate between the groups of patients. This methodology was applied respecting severe multicollinearity in the set of explanatory variables [20]. Statistical software SIMCA v. 12 from Umetrics (Umeå, Sweden) was used for the OPLS analysis [20].

## 3. Results

### 3.1. Clinical and Pathological Features

According to histology results, the patients (*n* = 178) were assigned to cohorts of benign (B) (*n* = 79; 44.4%; age median 55 (46–59) years), borderline (MB) (*n* = 10; 5.6%; age median 28 (27–47) years) and malignant (M) (*n* = 89; 50%; age median 42.5 (39–48) years) tumors. The B cohort patients were significantly older than the M and MB cohorts (*p* = 0.009). In the group MB, five non-invasive follicular thyroid neoplasms with papillary-like nuclear features (NIFTP) and five follicular tumors of uncertain malignant potential (FT-UMP) were included. The M cohort consisted mainly of PTC (93.3%); 66 patients had the classic variant of PTC, 11 had the follicular variant of PTC, one had the tall cell variant of PTC, and three papillary microcarcinomas (incidentalomas). Further, two poorly differentiated PTC, one anaplastic carcinoma, two follicular thyroid carcinomas (FTC), one oncocytic carcinoma with classic PTC, one metastasis of fibrous carcinoma, and one gastrointestinal stromal tumor (GIST) were identified.

Multinodular thyroid gland (MNTG) was present in 50% (*n* = 89) of patients. The presence of AITD was confirmed in 46.1% of patients (*n* = 82). The time range between the first clinical examination and the final histology was 7.6 years (1–240 months; *n* = 15) in the B cohort and 4 years (1–180 months; *n* = 17) in the M cohort; we only included patients with long-term documentation. The reasons for thyroid examination were as follows: (1) neck resistance detected by the patients, which were significantly more common in the M cohort (*n* = 19) in comparison to the B cohort (*n* = 10) (*p* = 0.035); (2) neck resistance detected by the general physician during preventive examination in M (*n* = 9) and in B (*n* = 6) (*p* = 0.466) cohort; (3) hoarseness, present in two patients with PTC; (4) complaints of “globus sensation“ in four patients in the M and 12 in the B cohort (*p* = 0.012); (5) the presence of thyropathy in the patient/family history or thyroid cancer in the family history, which were not different between the B (*n* = 11) and M (*n* = 18) cohorts (*p* = 0.114); and (6) the detection of incidentalomas, which were similar in both groups: B (*n* = 7) and M (*n* = 6). In other patients (*n* = 74), the reasons for thyroid examination were unknown/not mentioned.

The B cohort had larger thyroid nodules in comparison to the M cohort (*p* = 0.033). TSH levels were lower in the B cohort compared to the M cohort (*p* = 0.015). Anti-Tg levels were significantly higher in the M cohort (*p* = 0.007). For more detailed characteristics, see Table 1. As for sex differences, men had more commonly presented lymphadenopathy (*p* = 0.014), minimally invasive carcinoma (*p* = 0.002), positive TERT mutation (*p* < 0.01) and higher TNM (8th edition) staging of TC (*p* < 0.01) in comparison to women. Patients with TC had more commonly neck lymphadenopathy (*p* < 0.01) and AITD (*p* = 0.04) compared to benign histology (data not shown).

### 3.2. Risk Stratification of Thyroid Nodules on Ultrasound

The results of the risk stratification of thyroid nodules on ultrasound ACR TI-RADS are summarized in Figure 2. Additionally, the types of missed carcinomas with implemented molecular results are noted because FNA indications were not strictly followed according to the suggested ACR TI-RADS thyroid nodule size limits. If followed, we would have missed 18% of PTC. The interobserver agreement was already set up in one of our previous studies. Two experienced endocrinologists independently examined and rated US images available; the absolute difference was 5.95%, and individual SD was 4.20 [21].

### 3.3. Fine-Needle Aspiration Cytology and Molecular Testing

FNAC Bethesda findings were Bethesda II (*n* = 18; 10.1%), III (*n* = 52; 29.2%), IV (*n* = 36; 20.3%), V (*n* = 39; 21.9%), and VI (*n* = 33; 18. *n* 5%). The total detection rate of mutations was 8/79 (10.1%) in benign, 1/10 (10%) in MB, and 69/89 (77.5%) in malignant histologically proven specimens; additionally, six more mutations were revealed in surgically resected tissues; four fusion genes, one RAS + TERT mutation and one *BRAF* mutation. More details are given in Figure 3.

Patients with BRAF + TERT mutations were the oldest (72.3 (65–82) years), BRAF positive patients (BRAF V600E (*n* = 48); BRAF K601E (*n* = 1)) had an average age of 36.1 (16–46) years, and fusion-positive patients were 35.5 (12–60) years old. Only two FNA molecular tests were limited because of material quality. Additionally, two more patients had negative FNA molecular testing in contrast to positive molecular testing in histology, 1 NRAS Q61R + TERT C228T mutation in FTC and 1 BRAF V600E mutation in Bethesda V thyroid nodule with adequate FNA material; both patients had large thyroid nodules of 34 mL and 5.8 mL. TERT positive patients had the largest thyroid nodules of 16.2 (1.9–27.2) mL, fusions 3.8 (0.3–22.2) mL, and BRAF positive 2.3 (0.3–12.2) mL. Total thyroidectomies were performed in 75 patients in the M cohort, and additionally, total thyroidectomy with lymphadenectomy was conducted in 12 patients; only two patients underwent hemithyroidectomy. In the benign cohort of patients, total thyroidectomy was performed in 46 patients and hemithyroidectomy in 30 patients. In other patients (*n* = 13), the extent of surgery was unknown.

### 3.4. Thyroid Cancer Detection

The strongest relevant positive predictors for malignancy (t-statistic = 20.06; *p* < 0.01) with an explained variability 60.7% were the presence of the genetic mutations BRAF (14.60; *p* < 0.01), TERT (2.41; *p* < 0.05), gene fusions (2.83; *p* < 0.05), FNAC (11.01; *p* < 0.01), ACR TI-RADS (7.43; *p* < 0.01), and the presence of AITD (2.47; *p* < 0.05). The negative relevant predictor for malignancy was thyroid nodule size (−2.03; *p* < 0.05). For more details, see Table 2. The strongest relevant positive predictors for malignancy separately for the cohort of indeterminate FNA category Bethesda III + IV with an explained variability of 19.5% were the RAS mutation (t-statistic = 5.25; *p* < 0.01), ACR TI-RADS (4.08; *p* < 0.05), presence of AITD (3.93; *p* < 0.05), and MNTG (t-statistic = 1.38; *p* < 0.05) (data not shown).

FNAC and ACR TI-RADS then underwent ROC analysis. FNAC had a very good performance, especially for the category Bethesda V and VI, reaching a sensitivity of 70.8% (60.2, 80), specificity of 91% (83.1, 96), and AUC of 0.811 (0.735, 0.868) in comparison to ACR TI-RADS, with sufficient performance mainly for the TR5 group reaching a sensitivity of 39.7% (27.6, 52.8), specificity of 90% (81.2, 95.6), and AUC of 0.682 (0.595, 0.753) (Figure 4).

When combined, FNAC, ACR TI-RADS, and genetic testing reached a sensitivity of 96.1% (95% CI 88.9–99.2) and specificity of 88.3% (80.0–94.0), with a diagnostic odds ratio (DOR) of 183.6. Sensitivity, specificity, and DOR decreased to 75.0% (47.6–92.7), 88.9% (79.3–95.1), and DOR 24.0 separately for the indeterminate FNAC categories. For more details, see Table 3.

## 4. Discussion

The prevalence of thyroid nodules is very high in the general population, and of the vast number of thyroid nodules detected, only 5–15% are demonstrated to be thyroid cancer [12]. It is, therefore, crucial to easily distinguish between benign and malignant thyroid nodules. This would allow treatment strategy recommendations that are tailored to individual patients. In our prospective cohort study, we attempted to identify the optimal TC detection in routine clinical practice according to:(1)Basic clinical examination. In accordance with other studies, we support the conclusions of natural history studies demonstrating the indolent behavior of some thyroid tumors [22,23]. On the other hand, we are aware of tumors with aggressive phenotypes, and these should not be underestimated [24]. Generally, thyroid cancer screening in adults is not recommended by the US Preventive Services Task Force, apart from hoarseness, neck pain and/or resistance, painful swallowing, radiotherapy in childhood, thyroid-associated genetic syndromes, or a family history of TC [25]. In our study, neck resistance and hoarseness were present in our M cohort. Further, the risk of malignancy was approximately 50% in thyroid incidentaloma, as supported by other studies. Advanced TC is common in this group of patients [26,27]. We observed significantly higher levels of TSH (but still in the normal range) in patients undergoing thyroid surgery with histologically proven malignancies in comparison to lower TSH levels in the benign group. The reasons for our benign surgery results included multinodular goiter, multinodular toxic goiter, toxic nodule, and Graves’ disease. It is well-known that toxic nodules confirmed by scintigraphy are rarely malignant [1]. This difference in TSH levels between B and M cohorts established by histology compared to the general population is evident, and thus, TSH levels have not generally been helpful in the prediction of TC risk. We also confirmed our previous study results that obesity and glucose disorders are not substantive risk factors for TC [28]. Finally, the M cohort had significantly higher levels of anti-Tg and the presence of AITD. There have been many studies on AITD and TC, and as in our study, the results have been inconsistent. We suggest that the positivity exclusively of anti-Tg is just a secondary response to cancer antigens and not a sign of AITD [1,29,30].(2)Thyroid nodule stratification by ultrasound. ACR TI-RADS has demonstrated the highest diagnostic performance, being significantly superior to ATA and some other systems [31,32]. ACR TI-RADS has reduced the number of biopsies of benign nodules by more than twice in comparison to other systems (52.9% for ACR TI-RADS and 21.9% for the ATA guidelines). The ACR TI-RADS criteria allow a reduction in the percentage of benign nodules that are biopsied, which also results in a lower number of malignant nodules that are biopsied. This is unavoidable because there are some TC without typical suspicious sonographic features. In the study of Middleton et al., 31.8% of malignant nodules with the use of the ACR TI-RADS would not have been recommended for biopsy [4]. In one of our previous studies, we would have missed 17.9% TC, which is the same as in this current study, 18% [21]. Therefore, we did not strictly follow the thyroid nodule size limits indicating FNA. Most of our “missed” TC were pT1, but also one pT2 and one pT1 with lymphatic node metastasis, and thus in most cases, low stage PTC without the most unfavorable molecular markers results. In our study, ACR TI-RADS performed sufficiently, and up to 75% of thyroid nodules were correctly classified with a specificity of 90%. We had a very high cancer prevalence of 37.5% in benign FNAC. Molecular testing was done only in suspicious thyroid nodules with benign cytology according to suspicious ACR TI-RADS classification or by examiner recommendation. Benign cytological nodules with final malignant or borderline histology were as follows: fibrous carcinoma, NIFTP, FT-UMP, and FTC. The goal of the ACR TI-RADS is to minimize the number of clinically significant cancers that are missed. Follow-up recommendations should result in subsequent detection of some cancers that otherwise would have been overlooked [15,16,32].(3)FNAC had a very good performance in our study, especially for Bethesda V and VI reaching sensitivity and specificity of 70.8% and 91%, respectively. On average, 81% of patients were properly classified by FNAC. The previous study confirmed that the false-negative rate for benign FNAC is low, at 3.2%. Therefore, the standard-of-care approach of using FNAC and Bethesda system reporting standards is accurate and rarely misses malignancy [33]. However, indeterminate FNAC results were present in 49.4% of patients in our study, compared to other studies with ranges from 6% to 55% [17,34]. At this point, FNAC could not be presently improved except by using FNA molecular testing.(4)FNA molecular testing of genetic alterations in TC can be helpful in several ways. The benefits can be in a preoperative diagnosis, the extent of surgery, estimation of the prognosis, and determining the appropriate treatment for the patients. To comply with the rule-in and rule-out strategies, the necessary NPV for rule-out tests should be >95%, while the ideal PPV should be >95% for rule-in strategies leading to more radical resections (total thyroidectomy) following the National Comprehensive Cancer Network guidelines (NCCN). The patients with indeterminate thyroid nodules undergo diagnostic rather than curative thyroid surgery because a minority (about 20%) of these nodules have been shown to be malignant at final histology [35]. Mutational testing just for *BRAF* in AUS/FLUS samples has not been sufficient due to high specificity but low sensitivity for TC detection. However, molecular analysis using comprehensive genetic panels offers a significantly higher sensitivity of 91.1–94.4% [8]. Positive testing for *BRAF* or *RET/PTC* mutation has been shown to be specific for a malignant outcome in 100% of cases, whereas *RAS* mutations have been detected in up to 48% of benign follicular adenomas, 57% of FTC, and 21% of PTC [36,37]. It has been demonstrated that PTCs with both *BRAF* and *TERT* promoter mutations show the most aggressive characteristics and affect the prognosis of the patients [38,39,40]. Further, *NTRK* fusion genes are also valuable diagnostic and prognostic markers. *NTRK* fusion-positive carcinomas have been associated with the presence of AITD and lymph node metastases. *NTRK1*-rearranged carcinomas have been more aggressive with multifocality than *NTRK3*-rearranged carcinomas. The factors affecting a patient’s prognosis have been identified as follows: tumor size, the presence of metastases, positivity for the *NTRK3* or *NTRK1* fusion gene, and a late mutation event (*TERT* or *TP53* mutation) [41]. In our study, cancer prevalence in the indeterminate cytology category was 24.4%; in the AUS/FLUS group of patients, 10 out of 11 were correctly detected with molecular testing, and malignancy was confirmed, while in contrast, 6 out of 35 patients underwent thyroid surgery for the reason of positive molecular testing with benign histology (4× *RAS*, 2× *PTEN*). Therefore, 69.2% of patients with category Bethesda III could avoid surgery according to negative molecular testing. In the category Bethesda IV, only four out of nine patients were correctly detected by molecular testing; however, the number of patients was very small. In the cohort of patients with indeterminate cytological results, we also observed a tendency to preferential *RAS* mutation detection, especially in multinodular thyroid. In general, up to 86% of the indeterminate cytology group could be properly identified by molecular testing. Including all Bethesda categories, up to 92% could be correctly defined by molecular testing. We would also like to note that our results continuously improved because of the expansion of molecular testing. In the beginning, some TC were missed due to limited molecular testing, further in patients with incidentaloma (all pT1a), oncocytic tumor, FTC, fibrous carcinoma (metastasis), and, lastly, in patients with poor-quality FNA material. Molecular diagnostics may have limits on these neoplasms. Finally, it is necessary to be aware that panel-negative results should not be considered evidence for a benign tumor.(5)Changes in approaches in routine clinical settings. First, during basic clinical examination, gender and the age of the patients are simple parameters and should be considered (men are more at-risk for advanced TC in correlation with *TERT* mutation positivity in comparison to women). In young patients < 40 years in comparison to older ≥ 60 years, it was shown that papillary microcarcinomas (PMCs) were most likely to enlarge or show clinical node metastases [42].

Further, incorporating ACR TI-RADS and/or other similar US guidance classification is helpful in the stratification of the highly prevalent thyroid nodules. It is also a simple training tool for more inexperienced physicians.

Molecular testing can be helpful even in the category Bethesda II with US highly suspicious features or with suspicion raised by examiners. Patients with indeterminate FNA cytology can be followed by active surveillance (a molecular diagnostic test in conjunction with clinical and ultrasound features) with a predicted risk of malignancy that is comparable to the rate seen in cytologically benign thyroid FNA (approximately ≤ 5%) [34]. In our indeterminate group of patients, the negative predictive value according to molecular testing was 92.3%. Under our conditions, reFNA in the category Bethesda III is not indicated in patients with positive molecular testing, and surgery is recommended. In contrast, these patients with negative molecular testing are candidates for active surveillance. We are aware that studies of long-term follow-up in these patients are lacking. All our patients in Bethesda category IV are recommended for surgery. In our categories Bethesda V and VI, molecular testing mostly just confirmed the malignancy already identified more simply by FNAC. However, molecular testing can provide additive information about the prognosis, course of the TC, extent of the surgery, treatment, and follow-up that can be tailored to the individual patient. It is likely that findings of unfavorable genetic markers, even in a small tumor, would suggest an early stage of a clinically relevant PTC [38,39]. On the other hand, it must be considered that patients present at different stages of TC, with more advanced diseases requiring more radical surgery and, conversely, less advanced diseases requiring less radical surgery to avoid overtreatment. In addition, molecular testing may provide additional information to hesitating patients who need to know the individual risk of malignancy.

The strengths of our study include the prospective design, the inclusive cohorts of patients without prior selection, and that the data were confirmed by histology as the gold standard. Further, we made extensive efforts to collect detailed patient characteristics. To the best of our knowledge, only a few other studies are available that have just partially dealt with similar issues [43,44,45]. Most of our “missed” TC, according to ACR TI-RADS, were confirmed as a low stage PTC without the most unfavorable molecular markers results. The limitation of the study is the limited number of thyroid nodules, which makes the generalization of results more difficult. In addition, patient survival and cost-effectiveness were not assessed in our study. However, we hope that these limitations are offset by the comprehensive nature of our results.

## 5. Conclusions

We suggest that FNA molecular testing has the potential to substantially and progressively improve the potential for thyroid malignancy detection and can improve the approaches to treatments in patients. However, clinical examination, FNAC, and the risk stratification of thyroid nodules on ultrasound remain relevant factors. In addition, a broader spectrum of molecular markers must be involved to make correct diagnoses in all patients in a routine clinical setting.

## Figures and Tables

**Figure 1 biomedicines-10-00954-f001:**
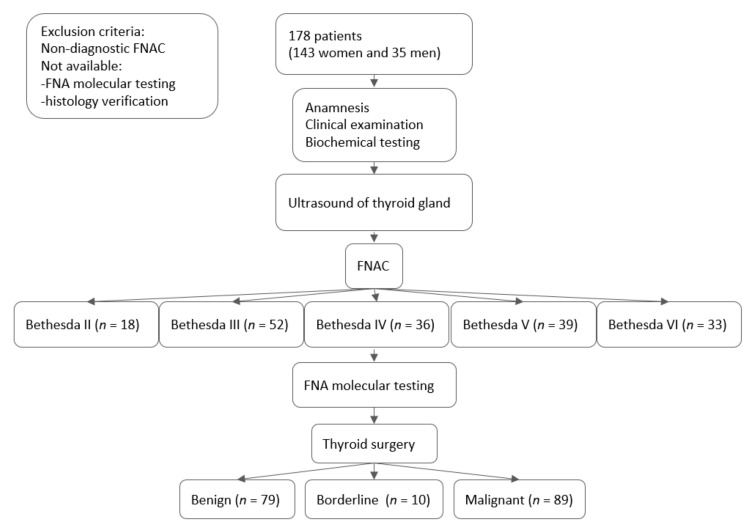
Flowchart of the study.

**Figure 2 biomedicines-10-00954-f002:**
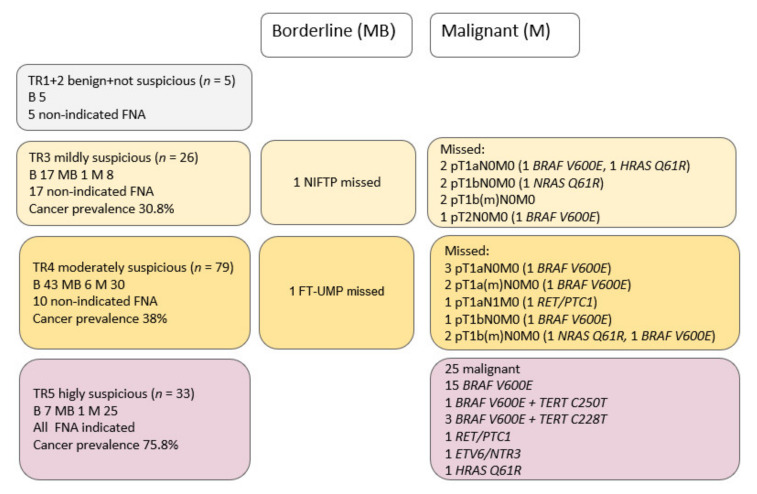
Risk stratification of thyroid nodules on ultrasound using the American College of Radiology Thyroid Imaging Reporting and Database System (ACR TI-RADS; TR) and staging (TNM 8th edition) of “missed’’ papillary thyroid carcinomas in TR3/TR4 groups if FNA indication would be strictly followed according to thyroid nodule size. Molecular genetic results are mentioned according to the ultrasound group TR1–TR5. Cancer prevalence was calculated as the number of people with TC in each group of patients according to ACR TI-RADS (excluding MB). Benign (B),borderline (MB), and malignant (M) cohorts of patients were determined by histology. NIFTP—non-invasive follicular thyroid neoplasm with papillary-like nuclear features; FT-UMP—follicular tumor of uncertain malignant potential.

**Figure 3 biomedicines-10-00954-f003:**
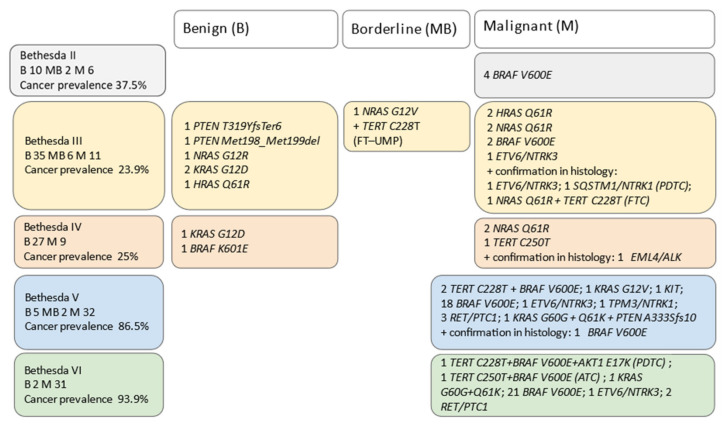
Bethesda System for Reporting Thyroid Cytopathology results with cancer prevalence in particular groups concurrently with molecular testing. Benign (B), borderline (MB), and malignant (M) cohorts of patients were determined by histology. M cohort consisted especially of PTC; other tumor types are pointed out. Cancer prevalence was calculated as the number of people with TC in each group of patients according to Bethesda categories (excluding MB).FT-UMP—follicular tumor of uncertain malignant potential; ATC—anaplastic thyroid cancer; PDTC—poorly differentiated thyroid cancer and FTC follicular thyroid cancer.

**Figure 4 biomedicines-10-00954-f004:**
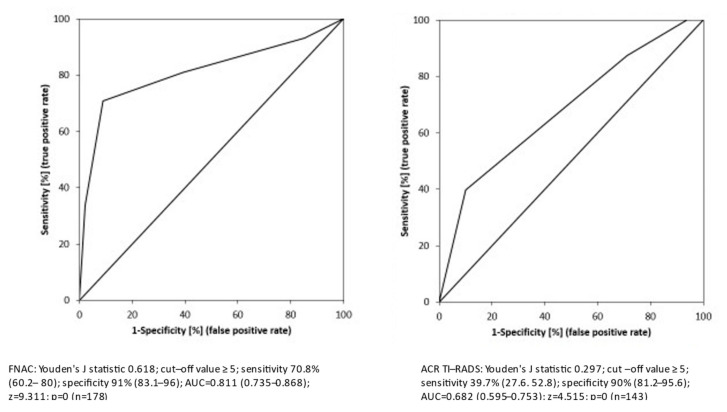
ROC analysis of FNAC and ACR TI-RADS, cut-off values ≥ 5 both FNAC (Bethesda category V–VI) and ACR TI-RADS (TR5).

**Table 1 biomedicines-10-00954-t001:** Comparison of clinical, biochemical, and imaging characteristics between B (benign), M (malignant), and MB (borderline tumor) cohorts of patients with Kruskal–Wallis one-way ANOVA on ranks (KW).

	B		M		MB		KW BxMxMB	KW BxM
Variable	Count	Median (CI 95%)	Count	Median (CI 95%)	Count	Median (CI 95%)		
Female	63		70		10			
Male	16		19		0			
Age (years)	79	55 (46–59)	89	42.5 (39–48)	10	28 (27–47)	0.001 B vs.M; B vs. MB	0.009
Thyroid Nodule (mL)	79	2.55 (1.4–3.8)	89	1.40 (1–2)	10	1.15 (0.7–5.0)	0.087	0.033
Thyroid Gland (mL)	79	16.90 (14–19.5)	89	14.50 (11.6–16)	10	16.1 (12.5–19.01)	0.179	0.07
TSH	58	1.35 (1.08–1.98)	58	2.03 (1.81–2.44)	10	0.92 (0.54–2.1)	0.014 B vs. M	0.015
fT4	57	15.80 (15–16.5)	58	15.30 (14.5–16.2)	10	16.65 (12.50–17.30)	0.525	0.245
anti–TPO	37	7.39 (4.34–12.3)	40	6.10 (2.53–18)	3	3.28	0.739	0.46
anti–Tg	39	3.55 (1.27–7.93)	40	10.91 (6.08–15.42)	3	3.53	0.023 B vs. M	0.007
Glycemia (mmol/L)	23	5.32 (5.2–5.7)	28	5.30 (5.1–5.5)	6	5.05 (4.6–5.33)	0.044 B vs. MB	0.255
BMI	40	26.90 (22.3–29.3)	38	27.10 (24.09–28.08)	4	22.8	0.235	0.586

**Table 2 biomedicines-10-00954-t002:** Relationships between the cohort of patients with thyroid cancer (M) and predictors for the 1st predictive component as evaluated by the O2PLS model and multiple regression (for details, see statistical analysis). Ra-component loadings are expressed as correlation coefficients with predictive components. * *p* < 0.05; ** *p* < 0.01. AITD—autoimmune thyroid disease.

**OPLS Predictive Component**						**Multiple Regression**			
	Variable	Component Loading	t‐Statistics	R*a*		Regression Coefficient	t‐Statistics		
Relevant Predictors (matrix **X**)	BRAF	0.567	14.6	0.787	**	0.326	24.05	**	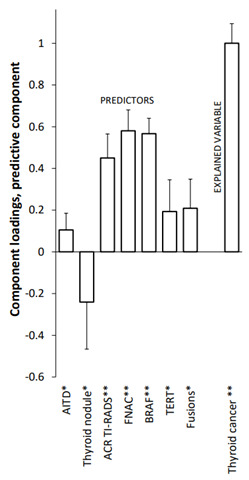
	TERT	0.194	2.41	0.261	*	0.086	3.08	**	
	Fusions	0.209	2.83	0.282	*	0.152	4.18	**	
	AITD	0.105	2.47	0.132	*	0.091	3.67	**	
	Thyroid Nodule	−0.241	−2.03	0.332	*	−0.125	2.23	*	
	FNAC	0.581	11.01	0.804	**	0.323	9.42	**	
	ACR TI–RADS	0.45	7.43	0.613	**	0.227	8.46	**	
(matrix **Y**)	**M**	1	20.06	0.779	**				
**Explained Variability**			60.7% (59.5% after cross–validation)						

**Table 3 biomedicines-10-00954-t003:** Performance of all methods ACR TI-RADS, FNAC, and molecular testing in thyroid cancer detection. The data were calculated for all cytological cohorts and separately for the cohorts of categories Bethesda III and IV. Positive predictive value (PPV), positive likelihood ratio (PLR), negative predictive value (NPV), negative likelihood ratio (NLR), and diagnostic odds ratio (DOR). Bethesda III–IV * and Bethesda II–VI * were calculated after including positive molecular testing additionally confirmed in histology (not in FNA).

	**Sensitivity (CI)%**	**Specificity (CI)%**	**PLR (CI)**	**PPV (CI)%**	**NLR (CI)**	**NPV (CI)%**	**Accuracy (%)**	**DOR**
Bethesda III+IV	66.7	84.2	4.22	55.8	0.4	89.4	80.2	10.6
	(39.1–86.2)	(74.4–90.7)	(2.19–8.13)	(39.6–70.8)	(0.18–0.89)	(79.1–95.0)	(70.3–87.9)	
Bethesda III–IV *	75	88.9	6.75	66.9	0.28	92.3	85.7	24
	(47.6–92.7)	(79.3–95.1)	(3.31–13.76)	(49.7–80.4)	(0.12–0.66)	(83.5–96.5)	(76.6–92.3)	
Bethesda II–VI	95.8	83.8	5.93	87	0.05	94.6	90.2	118.6
	(88.3–98.6)	(75.4–89.8)	(3.77–9.31)	(81–91.3)	(0.02–0.15)	(85.3–8.2)	(84.7–94.2)	
Bethesda II–VI *	96.1	88.3	8.21	90.3	0.04	95.2	92.4	183.6
	(88.9–99.2)	(80.0–94.0)	(4.70–14.33)	(84.1–94.2)	(0.01–0.14)	(86.7–98.4)	(87.3–95.9)	

## Data Availability

The data that support the findings of this study are available on request from the corresponding author.

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
