# Peer review of "Thyroid Cancer Detection in a Routine Clinical Setting: Performance of ACR TI-RADS, FNAC, and Molecular Testing in Prospective Cohort Study"

_biomedicines, 2022, doi:10.3390/biomedicines10050954_

Round 1

Reviewer 1 Report

In this work, Grimmichova and collaborators report the study of a cohort comprising 178 patients affected by thyroid nodules. The aim of the study was to compare the positive predictive value for malignancy of different diagnostic tests: ACR TI-RADS, FNAC and molecular testing.  The study is interesting although the results are not very novel. The utility of molecular testing in improving malignancy detection, especially in indeterminate FNAC cases is widely reported in literature. However, the study is well conducted and the results are significant.

I have only some suggestions to improve the strength of the study:

  1. Materials and Methods section should be splitted in sub-sections
  2. The Authors may include a Supplementary Table reporting all the characteristics of analyzed patients
  3. In Diagram 1, I found a bit confusing to report only the missed patients in the TR3 and TR4 groups and all the mutated patients in TR5 group
  4. In general, the contribution of molecular testing to diagnostic power may be better underlined. E.g. by adding a ROC curve taking in consideration also molecular testing. Also in Table 3, it is not very clear the contribution of molecular testing

Author Response

I am grateful to the reviewers for their insightful comments on my paper. I have been able to incorporate some changes to reflect the suggestions provided by the reviewers. I have highlighted the changes within the manuscript.

  1. Materials and Methods section should be splitted in sub-sections

    Response: I made the subsections. 

  1. The Authors may include a Supplementary Table reporting all the characteristics of analyzed    

    patients.

   Response: I put just the basic clinical characteristics in Table 1, I added the count of females, males. We made a more extensive analysis, but we pointed out just the significant ones in Results 3.1., especially line 231-235, because of just a few significance.

  1. In Diagram 1, I found a bit confusing to report only the missed patients in the TR3 and TR4 groups and all the mutated patients in TR5 group

Response: ACR TI-RADS has in general high specificity and lower sensitivity, up to 30% of malignant thyroid nodules can be missed, but I want to show what kind of tumors are missed, small ones without the most unfavorable molecular testing (for example TERT) as can be seen in group TR5. I tried to point it out in the discussion line 372-374.

  1. In general, the contribution of molecular testing to diagnostic power may be better underlined. E.g. by adding a ROC curve taking in consideration also molecular testing. Also in Table 3, it is not very clear the contribution of molecular testing

Response: The performance of single mutations are mentioned in Table 2. In comparison to ACR TI-RADS, FNAC etc, BRAF alone was performing the best. I´m sorry we did not make a ROC analysis for molecular testing, because of many mutations presenting different risk of cancer, for example RAS compared to TERT.

Table 3. ACR TI-RADS, FNAC and molecular testing all together, the effect of molecular testing can be seen indirectly just by adding more molecular tests. 

Reviewer 2 Report

The manuscript is very interesting and well written. The idea of the authors is brilliant and the results of clinical interest.

Only some minor points to clarify/add:

  • the title doesn't seem the right choiche. Thyroid nodule in a routine clinical setting: how to detect malignancy? for example, or something like that
  • nothing about thyroid scintigraphy was written or analyzed. This is a crucial point. Many thyroid nodules are studied with thyroid scan...also this point needs to be discuss and investigate in the text.
  • In table 1 gender is lacking
  • instead of diagram, use the term figure

Author Response

 I am grateful to the reviewers for their insightful comments on my paper. I have been able to incorporate some changes to reflect the suggestions provided by the reviewers. I have highlighted the changes within the manuscript.

  1. the title doesn't seem the right choice. Thyroid nodule in a routine clinical setting: how to detect malignancy? for example, or something like that

   Response: yes, I changed the title.

  1. nothing about thyroid scintigraphy was written or analyzed. This is a crucial point. Many thyroid nodules are studied with thyroid scan...also this point needs to be discuss and investigate in the text.

Response: I mentioned a little bit in discussion line 347.  You are absolutely right about scintigraphy, but almost all patients with any type of thyroid disease are under care of the endocrinologist in my country. That's probably why we don't use scintigraphy so often, we should. Scintigraphy was not involved in our study design.

  1. In table 1 gender is lacking. 

Response: I added it. 

  1. instead of diagram, use the term figure

Response: yes, I changed it.